# Sub-surface deformation of individual fingerprint ridges during tactile interactions

**Giulia Corniani[1,2]\*[†], Zing S Lee[3], Matt J Carré[3], Roger Lewis[3], Benoit P Delhaye[4,5]\*, Hannes P Saal[1,2]\***

[1]Active Touch Laboratory, School of Psychology, University of Sheffield, Sheffield, United Kingdom; [2]Insigneo Institute for in silico Medicine, University of Sheffield, Sheffield, United Kingdom; [3]Human Interaction Group, School of Mechanical, Aerospace and Civil Engineering, University of Sheffield, Sheffield, United Kingdom; [4]Institute of Information and Communication Technologies, Electronics and Applied Mathematics, Université Catholique de Louvain, Louvain-la-Neuve, Belgium; [5]Institute of Neuroscience, Université Catholique de Louvain, Louvain-la-Neuve, Belgium

**\*For correspondence:**
gcorniani@mgh.harvard.edu (GC);
benoit.delhaye@uclouvain.be (BPD);
h.saal@sheffield.ac.uk (HPS)

**Present address:** [†]Harvard Medical School at Spaulding Rehabilitation Hospital, Boston, United States

**Competing interest:** The authors declare that no competing interests exist.

## eLife Assessment

By leveraging optical coherence tomography this study provides **important** insight into the deformation of human fingertip ridges when contacting raised features such as edges and contours. The study provides **compelling** evidence that such features tend to cause deformation and relative movement of what the authors term ridge flanks rather than bending of the ridges themselves.

**Abstract** The human fingertip can detect small tactile features with a spatial acuity roughly the width of a fingerprint ridge. However, how individual ridges deform under contact to support accurate and high-precision tactile feedback is currently unknown. The complex mechanical structure of the glabrous skin, composed of multiple layers and intricate morphology within which mechanoreceptors are embedded, makes this question challenging. Here, we used optical coherence tomography to image and track sub-surface deformations of hundreds of individual fingerprint ridges across ten participants and four individual contact events at high spatial resolution in vivo. We calculated strain patterns in both the stratum corneum and viable epidermis in response to a variety of passively applied tactile stimuli, including static indentation, stick-to-slip events, sliding of a flat surface in different directions, and interaction with small tactile features, such as edges and grooves. We found that ridges could stretch, compress, and undergo considerable shearing orthogonal to the skin surface, but there was limited horizontal shear. Therefore, it appears that the primary components of ridge deformation and, potentially, neural responses are deformations of the ridge flanks and their relative movement, rather than overall bending of the ridges themselves. We conclude that the local distribution of mechanoreceptors across the ridges might be ideally suited to extract the resulting strain gradients and that the fingertip skin may possess a higher mechanical spatial resolution than that of a single ridge.

## Introduction

The human fingertip is a highly effective tactile sensing structure housing thousands of mechano-receptors (*Corniani and Saal, 2020*) that convert skin deformations into neural responses (*Handler*

*and Ginty, 2021*), which underlie our capacity for fine tactile discrimination and object manipulation (*Lieber and Bensmaia, 2022*; *Johansson and Flanagan, 2009*; *Witney et al., 2004*). The ridged structure of the volar skin of the fingertip gives rise to fingerprints, which are thought to determine the spatial resolution at which we can resolve small tactile features. Ridge spacing correlates with perceptual spatial acuity (*Peters et al., 2009*), and the size of receptive sub-fields of type-1 tactile afferents has been estimated in the sub-millimeter range, roughly matching the width of a single ridge (*Jarocka et al., 2021*; *Sukumar et al., 2022*).

However, understanding how the mechanical properties of individual fingerprint ridges support high-precision tactile feedback presents significant challenges: any force applied to a papillary ridge on the skin's surface causes deformations within multiple skin layers with distinct mechanical properties and complex morphology (*Dahiya and Gori, 2010*), before being transduced into neural activity by mechanoreceptors situated at the epidermis-dermis border, which are situated at specific landmarks relative to the limiting and intermediate ridge structures of the viable epidermis (see illustration in *Figure 1A*).

While numerous studies have investigated ridge deformations (*Cauna, 1954*; *Gerling and Thomas, 2005*; *Gerling and Thomas, 2008*; *Willemet et al., 2021*; *Delhaye et al., 2016*) and the mechanical properties of different skin layers (*Pereira et al., 1991*; *Leyva-Mendivil et al., 2017*; *Leyva-Mendivil et al., 2015*), most of this work has focused on ex-vivo specimens, surface measurements, or computer simulations. Consequently, the mechanical response of the skin below its immediate surface remains largely unknown, leading to conflicting interpretations in the literature. For instance, it has been proposed that the papillary ridges are stiffer than the neighbouring grooves (*Swensson et al., 1998*), which might imply that normal loading of the skin might not affect the ridges' profile appreciably. Conversely, other observations have suggested that the grooves are relatively stiff, allowing the papillary ridges to deform considerably (*Cauna, 1954*; *Johansson and LaMotte, 1983*). However, the sub-surface consequences of this putative pliability during object contact or stick-to-slip transitions (see e.g. *Delhaye et al., 2016*) are unclear: the whole ridge structure might bend as proposed in Cauna's lever mechanism (*Cauna, 1954*), but this view has proved controversial (see e.g. *Gerling and Thomas, 2008*), with direct empirical evidence lacking.

From a mechanical standpoint, these conflicting interpretations raise the question of how the outermost two skin layers typically deform at the resolution of single papillary ridges, whether by tension, compression, or shear (see examples in *Figure 1B*). Additionally, such deformations might apply to individual papillary ridges and all their sub-surface structures equally, for example, horizontal shearing that bends the papillary ridge in a certain direction, while levering its sub-surface aspects in the opposite direction. Conversely, individual parts of the ridge structure might deform differently. For example, the viable epidermis might deform to a different extent or in a different direction due to its lower stiffness and different morphology. Additionally, if there are indeed mechanical differences between papillary ridges and their neighbouring grooves at the level of the stratum corneum, this might result in differential movements of the two sides of each papillary ridge, here referred to as ridge flanks (see *Figure 1B-iv*, right, for a potential example).

To empirically address these questions, we employed Optical Coherence Tomography (OCT) to precisely measure the sub-surface deformation of individual fingerprint ridges in response to a variety of mechanical events. Specifically, we focused on the stratum corneum and the bulk of the viable epidermis (excluding intermediate ridges), which could be robustly resolved and tracked by our setup. OCT has been used previously to investigate sub-surface skin properties, however, its application has been limited to static characterization of the skin morphology, such as measuring tissues' thickness and roughness (*Maiti et al., 2020*; *Ding et al., 2021*; *Adabi et al., 2017*; *Czekalla et al., 2019*; *Lin et al., 2021*). In the present study, we aimed to investigate the ways in which ridges are able to deform during dynamic contact events as well as the role played by different skin layers in order to better understand how the ridged structure of fingertip skin supports tactile sensing.

## Results

### Measuring skin deformation on a sub-ridge scale

We developed a setup for tactile stimulation of the fingertip, which involved lowering a custom-made transparent thin plate onto the participant's fixed fingertip at a set normal load and smoothly sliding

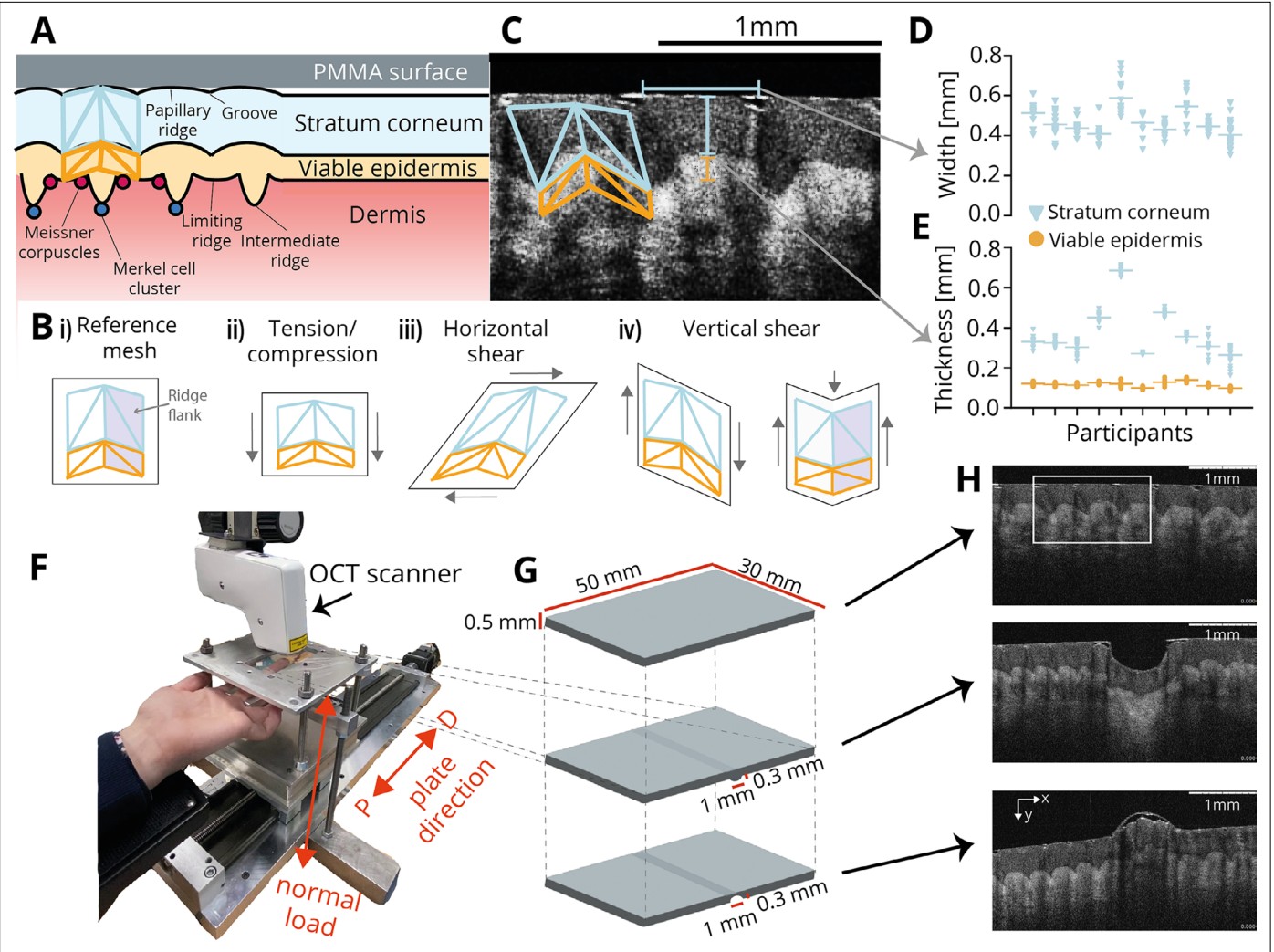

**Figure 1.** Experimental setup. (**A**) Two major mechanically relevant skin layers constitute the epidermis on the human fingertip: the stratum corneum (blue shading) and the viable epidermis (orange). The papillary ridges making up the fingerprints at the skin surface extend to deeper layers, where morphological complexity increases. Interactions with surfaces (gray shading) take place at the outer boundary of the stratum corneum, while mechanoreceptors are located at the border between the epidermis and the dermis (red shading), at distinct landmarks associated with the ridge structure (dark red and blue circles). Identification of morphological landmarks related to the ridge structure allows the creation of a fine-grained mesh covering most of the sub-surface structure of a single papillary ridge (blue and orange overlaid meshes). (**B**) Potential deformations of the tracked ridge structure, including the stratum corneum and the bulk of the viable epidermis, during tactile interactions, with arrows indicating the directions of relative deformation. (**i**) Reference mesh (undeformed); (**ii**) Tension or compression of the whole structure along or orthogonal to the surface axis of the skin; (**iii**) horizontal (surface) shear, where the ridge structure tilts sideways; (**iv**) vertical shear, where the ridge structure tilts along the axis orthogonal to the surface of the skin. These deformations might affect different parts separately, e.g., via shearing in different directions across both ridge flanks as shown on the far right (see darker shading to highlight a single ridge flank). (**C**) Detail view of a single Optical Coherence Tomography (OCT) frame showing ridged skin structure and clear boundary between the stratum corneum and viable epidermis. A mesh covering the stratum corneum and the upper part of the viable epidermis (without the intermediate ridge) is overlaid spanning a single papillary ridge. The border between the viable epidermis and dermis is less clearly delineated, but some deeper features are resolved less well. (**D**) Ridge widths ($n = 153$) across all participants. Each dot corresponds to a unique ridge with the participant mean indicated by the horizontal line. (**E**) Thickness of stratum corneum (blue) and viable epidermis (yellow). Markers, as in **D**. (**F**) The experimental apparatus consisted of a finger holder to which the left middle finger of the participant was secured. A horizontal plate with a smaller inlaid transparent surface could be moved to indent the fingertip. A motorized linear stage moved the plate in the distal/ proximal direction across the fingertip. An OCT scanner recorded images through the transparent surface, while forces were recorded using a 3-axis force sensor. (**G**) Illustration of the three transparent polymethyl methacrylate (PMMA) surfaces: a flat surface, one embossed with a rounded edge, and a grooved one. (**H**) Individual images recorded by the OCT scanner using the three surfaces shown in panel **G**.

The online version of this article includes the following video for figure 1:

**Figure 1—video 1.** Recordings of sub-surface skin deformations.

https://elifesciences.org/articles/93554/figures#fig1video1

the plate in both distal and proximal directions at a set speed (*Figure 1F and G*). Concurrently, we used an OCT scanner to capture images of the skin's surface and sub-surface morphology. The scanner obtained 4 mm wide slices with 4.5 µm lateral and 5 µm axial resolution at a sampling frequency of 10 Hz. The images were taken by scanning through the transparent plate along the proximal-distal axis of the finger. Both the stratum corneum and viable epidermis were clearly distinguishable from each other (*Figure 1C and H*).

After acquiring and pre-processing the OCT images, we used semi-automatic methods to track sub-surface landmarks associated with individual ridges across different video frames (see Methods, *Figure 1—video 1*). The resulting triangular mesh consisted of eight facets per ridge, distributed between the stratum corneum and viable epidermis, as well as the two flanks of the ridge (*Figure 1B*). Importantly, the mesh was fine-grained enough to identify canonical sub-surface ridge deformation patterns in response to various tactile events. The mesh also extended to and partially covered the dermis-epidermis border, where low-threshold mechanoreceptors are located (*Figure 1A*). We measured an average ridge width of 0.47 mm across participants (*Figure 1D*), consistent with previous studies (*Moore, 1989*; *Ohler and Cummins, 1942*). Average skin layer thickness was 0.38 mm for the stratum corneum and 0.12 mm for the viable epidermis across our dataset (*Figure 1E*), again in agreement with previous studies using both in vivo imaging and ex vivo histology (*Fruhstorfer et al., 2000*; *Lintzeri et al., 2022*; *Maiti et al., 2020*). We gathered a large and varied dataset to be able to make robust claims: across four protocols and ten participants, we identified a total of 393 unique ridge slices, which were tracked for 34 s on average and more than a minute in some cases, supported by hundreds of thousands of individually tracked landmarks.

## Static normal load

To investigate the response of individual ridges to static normal loads, we applied a flat stimulus to the fingertip, incrementally increasing the load from initial contact in steps of 0.5 N until reaching a maximum of 3.5 N (see examples in *Figure 2A and B*). Deformation was measured relative to the unloaded ridge's stereotypical configuration, and the principal tensile and compressive components of the strain experienced by each mesh facet were obtained (see Methods).

Under load, the central part of individual ridges depressed, while the flanks stayed in place (vertical shear), which led to a flattening of the ridge structure and which was consistently observed across participants (see examples in *Figure 2C*). Ridges could also display horizontal shear and bend proximally or distally depending on their unloaded configuration and sometimes stretch horizontally along the axis of the skin surface. Both tensile and compressive strains increased monotonically with normal load, with rapid changes and, therefore, larger deformations at initial contact followed by slower and mostly linear changes from 0.5 N onwards (*Figure 2D and E*). Across all facets in a given layer, tensile strains reached around 15% in the stratum corneum and 10% in the viable epidermis, while compressive strains were above 20% in the viable epidermis and steadily increased with the load but remained below 10% in the stratum corneum. We noted a small increase in the area of the stratum corneum, which was possibly an artifact due to the imperfect fit of the mesh to the ridge's curvature (but see Discussion for an alternative explanation). In contrast, we found a notable decrease in the area of the viable epidermis (*Figure 2F*). Our results, therefore, indicate that both the stratum corneum and the viable epidermis undergo deformation under static load, manifested predominantly as a flattening of the papillary ridge and its sub-surface structure, but that compression is higher in the viable epidermis.

## Lateral sliding and stick-to-slip

After characterizing the deformation resulting from a static normal load, we investigated sliding interactions between the fingertip and the flat surface. The flat surface was first lowered onto the fingertip until stable contact was established (with a normal load below 0.5 N), then displaced at a constant speed of 0.8 mm/s by 7.6 mm in the distal or proximal direction for a total of 8 movements. In accordance with previous findings, we observed two distinct phases of relative movement between the skin and the plate: stick, where the ridges adhered to the plate and were dragged along with it, and slip, where the ridges stopped moving, and the plate slipped over them. The clear distinction between the stick and slip phases allowed us to classify each phase based on the average horizontal velocity over all tracked features (see *Figure 3A* for an example from a single participant).

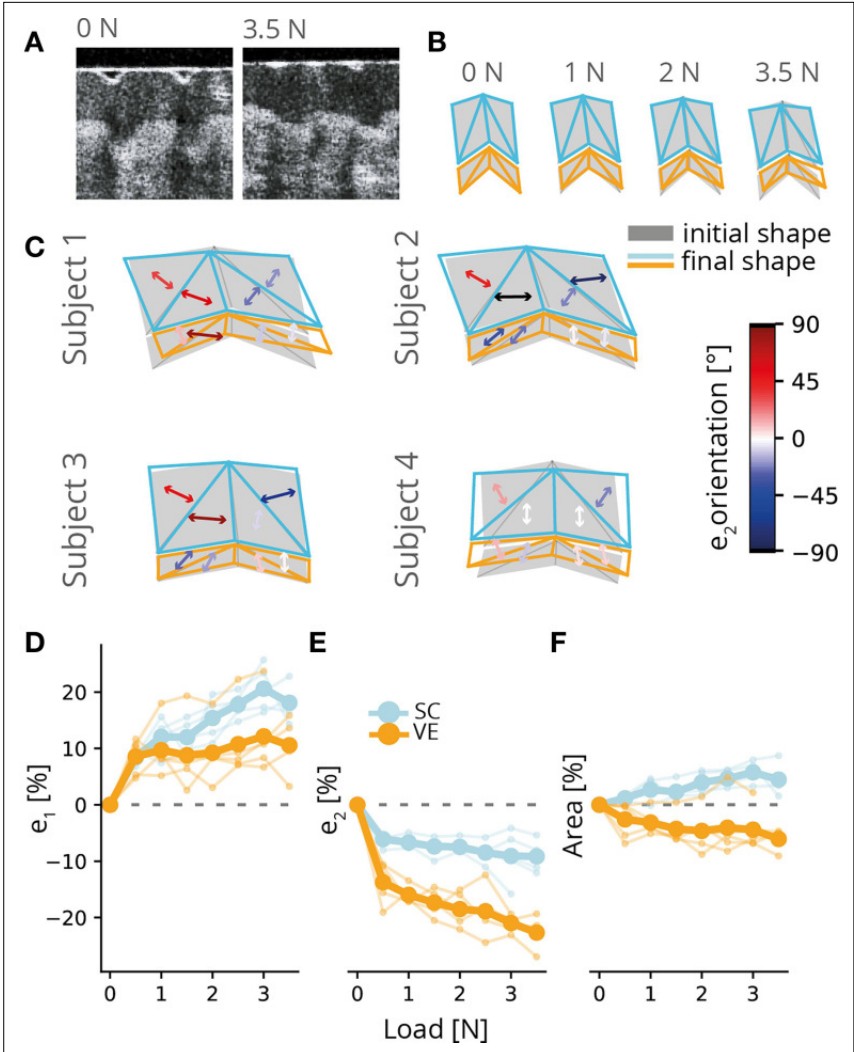

**Figure 2.** Ridge deformation under normal load. (**A**) Example frames showing a single ridge unloaded (0 N; left) and maximally loaded (3.5 N; right). (**B**) Examples of ridge deformation under static normal load for different load conditions. Shown is the unloaded ridge outline (gray shaded regions) with the deformed ridge mesh superimposed (blue and orange), under 1 N, 2 N, and 3.5 N loading. (**C**) Examples of ridge deformation under static normal load for different participants. Shown is the unloaded ridge outline (gray shaded regions) with the ridge mesh under 3.5 N loading superimposed. Arrows indicate the principal compressive axis for each facet, with colors indicating different orientations relative to vertical. In the lower panels, the magnitude of principal tensile ($e_1$, **D**) and compressive ($e_2$, **E**) strains as well as area change ($e_a$, **F**) as a function of normal load. Thin lines denote individual participants, and thick lines show the average. Data from five participants with seven individual ridges tracked each.

Previous studies have shown that stick-to-slip events elicit strain waves on the surface of the fingertip skin (*Delhaye et al., 2021*; *Willemet et al., 2022*; *du Bois de Dunilac et al., 2023*), but their effects on the sub-surface structure are unclear. Additionally, reversing the plate's movement direction changes the tangential force applied to the skin and induces shear deformations within the tissue, but it is unclear whether these are absorbed in surface skin layers or in deeper tissues. To quantify the deformation that occurred between the transitions from stick to slip and between different movement directions of the plate, respectively, we calculated a stereotypical mesh covering a single papillary ridge by averaging the meshes of all tracked ridges ($n = 140$ in total) across image frames within the same phase for each participant (*Figure 3B and C*, see Methods). We then calculated the principal tensile and compressive strain components experienced by each mesh facet on the transition from one phase to the next.

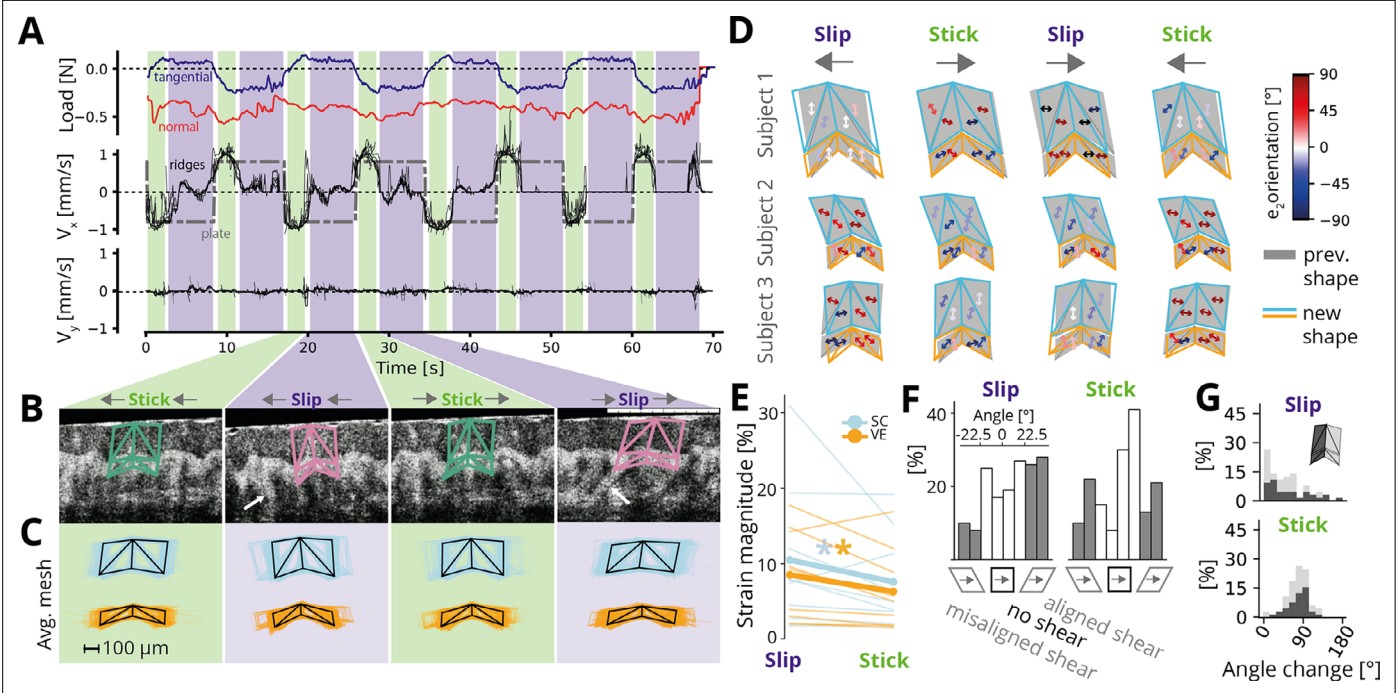

**Figure 3.** Ridge deformations during sliding. (**A**) Measurements during repeated movements of the flat plate along the distal-proximal axis for a single participant. Top: Tangential (purple) and normal (red) load as a function of time. The normal load was set at 0.5 N at the start of the trial by adjusting the indentation and then was not further controlled. Tangential load alternates between positive and negative values depending on the movement direction of the plate. Middle: Horizontal velocity of the plate (dash-dotted gray line) and average velocity of each tracked fingerprint ridge (thin black lines) during all eight transitions of the flat plate. Two phases are evident: when the ridge is moving along with the plate (stick, indicated by green shading) and when the ridge is stationary, but the plate is moving (slip: indicated by pink shading). Bottom: Vertical velocity of all tracked ridges along the normal axis, which is close to zero. (**B**) Example frames showing a single-tracked ridge for the stick and slip phases of the two movement directions. White arrows point to presumed collagen fiber bundles anchoring the skin to the bone. (**C**) Average ridge meshes (black lines) during each of the four phases calculated over all tracked ridges and all time points assigned to each phase for the same participant as in **A**, **B**. Colored lines indicate individual sample meshes (*n* = 3773; blue: stratum corneum, yellow: viable epidermis). (**D**) Examples showing ridge deformation across the four phases for three participants. Shown are the ridge outline for the previous phase (gray shaded regions) with the ridge mesh for the current mesh superimposed. Arrows indicate the principal compressive axis for each facet, with colors indicating different orientations relative to vertical. (**E**) Average magnitude of maximal shear strains as a proxy for overall deformation in the stick-to-slip transition compared to the movement reversal. Thin lines denote individual participants (averaged over facets and movement directions), and thick lines show the grand average. Asterisks denote statistically significant differences (paired Wilcoxon tests). Strains are about a third higher during stick-to-slip transitions than during movement reversals. (**F**) Histograms of average principal strain angles for all mesh facets and participants in stick-to-slip transitions (left) and movement reversals (right). White bars denote angles that are within 22.5 degrees of the coordinate axes (horizontal or vertical) and, therefore, denote tension or compression without considerable shear. Gray bars denote angles within 22.5 degrees of the diagonal and, therefore, denote horizontal shear. Positive angles denote shear acting in the same direction as the plate movement, while negative angles denote shear in the direction opposite to the plate movement. (**G**) Change in principal strain angles when transitioning to slip (top) or stick (bottom) phases, separated by ridge flank (light and dark gray). Movement reversals cause a 90 degree shift in the strain angle, while stick-to-slip transitions cause little change, with no differences between ridge flanks evident.

The online version of this article includes the following figure supplement(s) for figure 3:

**Figure supplement 1.** Ridge deformations for all participants during sliding of the flat plate.

Examining the ridge deformation changes between movement phases revealed large variability across participants, both in the magnitude and the orientation of the deformations (see individual examples in *Figure 3D* and all participants in *Figure 3—figure supplement 1*). This variability is likely explained by the fact that, during stick-to-slip transitions, some parts of the fingertip experience tension, while others experience compression (*Delhaye et al., 2016*), and, therefore, measurements depend on the precise location. Additionally, the mechanical properties of each individual fingertip influence its own mechanical response, and crucially, the timing of the stick-to-slip transition during plate transit (see Methods), leading to varied mechanical environments. Nonetheless, several consistent findings emerged from the data.

First, we found that, across participants, strain magnitudes (quantified as maximal shear strain, see Methods) were consistently larger by about a third during the stick-to-slip transitions compared to the plate reversal transition (*Figure 3E*) for both the stratum corneum ($p < 0.01$, paired Wilcoxon signed-rank test across all facets, movement directions, and participants) and viable epidermis ($p < 0.01$). This was also true for individual participants, with strain largest during stick-to-slip transitions for eight out of nine participants. Thus, ridge deformations are most pronounced during stick-to-slip transitions rather than in response to different movement directions of the surface.

Second, while the orientation of the principal strain axes varied between participants, they were relatively consistent within participants in individual phases: the average angular standard deviation was 26 degrees when considering all facets of the ridge within a given phase and, subsequently, averaging over phases and participants. Consequently, strains tended to act in a consistent direction along the entire extent of the ridge (see also examples in *Figure 3D*). Notably, the principal strain directions exhibited a greater alignment with the cardinal axes (0 and ±90 degrees) rather than diagonal axes (±45 degrees) for the majority of participants across most phases (averaging 60% overall, see *Figure 3F*). This alignment implies a predominance of tension and compression in the ridge rather than horizontal shear. When shear did manifest, it was more pronounced during stick-to-slip transitions compared to direction changes, and generally aligned with the direction of plate movement, although this alignment was not always observed. We did not find any significant differences in the distribution of principal angles between the stratum corneum and the viable epidermis for either stick-to-slip transitions or movement reversals ($p > 0.37$ for both based on two-sample Kolmogorov-Smirnov tests across all facets, directions, and participants).

The low amount of shear was surprising, as during stick phases, the skin was typically dragged along with the surface for several millimeters (mean: 3.4 mm, range: 0.9–6.3 mm). Assuming no motion of the bone and a distance of 5–10 mm between the lightly loaded skin surface and the bone (*Birznieks et al., 2001*) yields large expected shear strains of 45% on average (range: 9–126%), clearly much in excess of what we observed. Thus, it is possible that shearing is sustained by deeper tissues, an effect that could be tested in future studies by directly tracking the angle and orientation of collagen fiber bundles anchoring the epidermis to deeper tissues (see highlighted examples in *Figure 3B*).

Finally, when switching movement direction, which caused the skin to stick to the surface again, we observed a 90 degree rotation of the principal axis of deformation (see *Figure 3G*, bottom panel). In contrast, there was relatively little change in the angle when transitioning to slip (*Figure 3G*, top panel). Thus, changing direction caused a reversal in the tension and compression axes but relatively little deformation in itself; this deformation then occurred during the transition to slip. The observed changes in orientation occurred consistently for both ridge flanks and, therefore, applied uniformly to the whole ridge.

In summary, our findings indicate that the primary deformations occurring during contact with a flat plate predominantly involves tension and compression of individual ridges along the skin's surface. Horizontal shear is only occasionally present and does not consistently align with the direction of movement. When shear is observed, it is more prominent during stick-to-slip transitions rather than during reversals of the plate movement.

## Ridge interactions with small tactile features

Finally, we investigated ridge deformation during contact with small tactile features that were close in size to that of a single ridge itself (1 mm in width and 0.3 mm in height/depth). For this experiment, we used the plates with the edge and groove, respectively, keeping all other stimulation parameters the same. We identified five consecutive phases for each tracked ridge based on its position relative to the edge or groove: (1) sliding under the flat part of the surface, (2) approach of the feature, (3) located centrally under the feature, (4) withdrawal from the feature, and (5) sliding under the flat part of the surface (see illustrations in top rows of *Figure 4A and D*). We again quantified the deformation of the average ridge during all these phases (see Methods), with the strains calculated relative to the initial mesh before contact with the feature was made (phase 1).

Ridge deformations were remarkably similar across participants but differed clearly across movement phases (see ridge deformations for a single participant in *Figure 4A and D* bottom rows and all participants in *Figure 4—figure supplements 1 and 2*). There was minimal evidence supporting the notion of a rigid ridge maintaining its shape, and instead, the central part of the ridge deformed

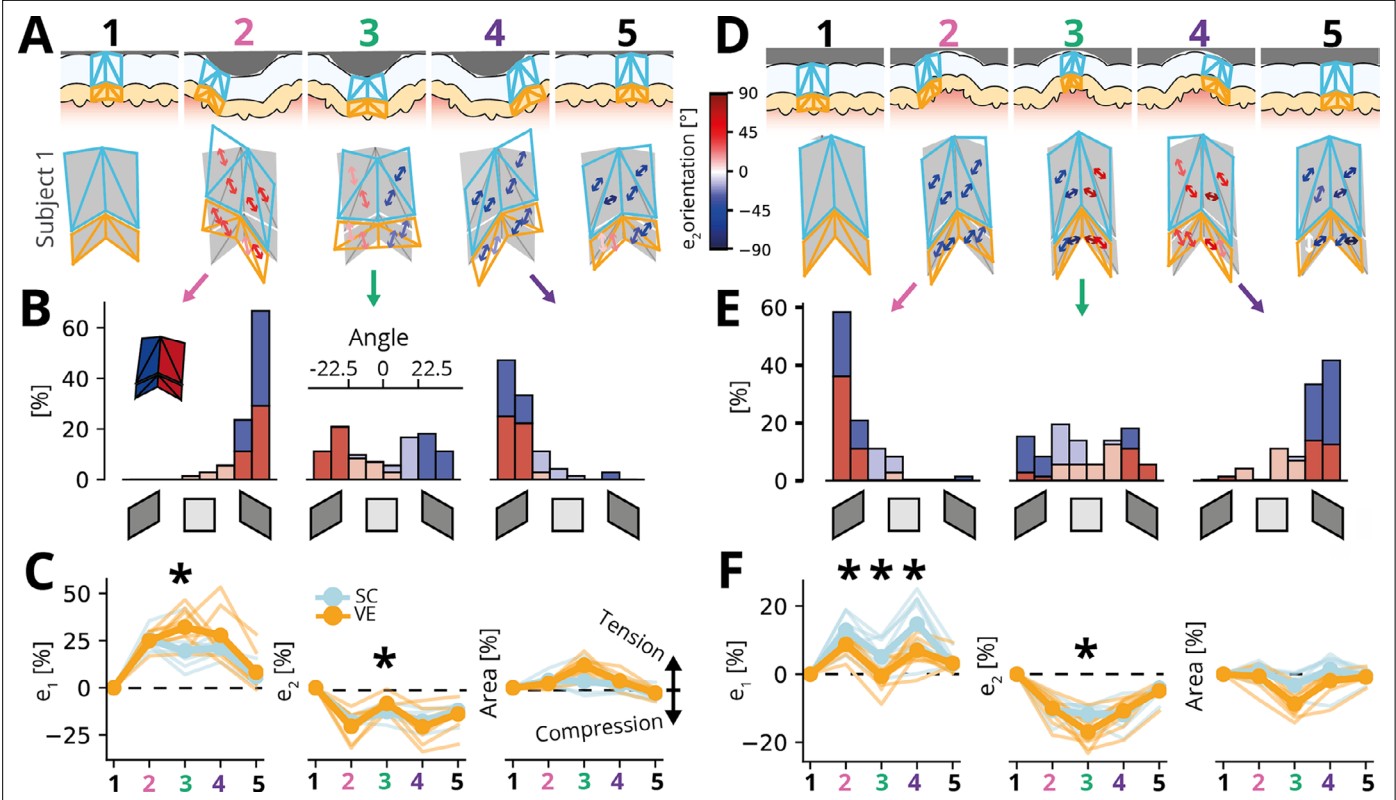

**Figure 4.** Ridge deformations and skin strains during transit of small tactile features. (**A**) Top row: Illustration of identified movement phases with the edge feature in different locations relative to the tracked fingerprint ridge. Bottom row: Average ridge deformation and associated principal compressive strain orientation for a single participant. Note that all strains are calculated with respect to the mesh shown in the left-most column, which represents the ridge during full slip before the interaction with the tactile feature. (**B**) Histograms of principal strain orientations across all ridge facets and participants for the approach (2), central (3), and withdrawal (4) phases. Red bars denote distal ridge flanks, and blue bars denote proximal ones. Darker shading denotes orientations close to diagonal, indicating shear, while lighter shadings denote angles aligned vertically or horizontally and, therefore, denote no shear. (**C**) Average magnitude of principal tensile ($e_1$, left panel) and compressive ($e_2$, middle panel) strains as well as area change (right panel) relative to the stereotypical ridge under full sliding (phase 1). Thin lines denote individual participants ($n = 9$), and thick lines show the average. Asterisks denote statistically significant differences between the stratum corneum and the viable epidermis (Bonferroni corrected paired Wilcoxon tests). Ridges are most deformed when directly under the edge feature when strains are higher in the viable epidermis than the stratum corneum. While the stratum corneum is incompressible, the area of the viable epidermis expands when directly under the ridge. (**D–F**) Same as in **A–C**, but for a small groove transitioning over the ridge. Note that the findings broadly reflect those obtained with the edge feature, but with the direction of compressive and tensile strains reversed.

The online version of this article includes the following figure supplement(s) for figure 4:

**Figure supplement 1.** Ridge deformations for all participants during transit of the edge feature.

**Figure supplement 2.** Ridge deformations for all participants during transit of the grooved feature.

considerably. Horizontal shear was relatively modest, with generally little bending of the ridge observed. Instead, our findings highlighted that the primary mechanism through which the ridge conformed to the shape of a feature involved vertical shearing of the whole ridge or the two ridge flanks against each other, as the skin moved vertically to conform to the feature. Consequently, during the transit of the edge, the full ridge initially sheared orthogonally to the skin's surface with both the strain orientations of both flanks aligned (*Figure 4B*, left panel). When the edge was directly above the ridge, both flanks sheared against each other, leading to opposite strain orientations across the two flanks (*Figure 4B*, middle panel) before the ridge returned to its original configuration during the withdrawal of the edge (*Figure 4B*, right panel). This pattern was mirrored, but with strain orientations reversed, during the transit of the groove (*Figure 4E*).

Strains increased and reached their maximum with respect to the initial configuration when the ridge was located directly under the feature and then decreased again as the ridge reverted to its original shape. For the edge, both strain components differed between the viable epidermis compared

to the stratum corneum when the ridge was located directly below the feature ($p < 0.01$ in Bonferroni corrected paired Wilcoxon tests across all facets and participants), resulting in an expansion of the area of the viable epidermis, as it was stretched around the edge feature, while the mostly incompressible stratum corneum retained its area (*Figure 4C*). Conversely, while interacting with the groove, a ridge experienced high compressive strains, which again were larger in the viable epidermis compared to the stratum corneum, and as a result, the area covered by the viable epidermis contracted, while the stratum corneum retained its area (*Figure 4F*).

## Discussion

This study measured the sub-surface deformation of individual papillary ridges in the fingertip skin during tactile interactions in vivo. The dataset included hundreds of individual ridges, whose deformations were tracked using high-resolution meshes during static normal load, lateral stick, and slip using a flat surface, as well as during contact with small tactile features. We found that individual ridges tended to flatten considerably at relatively low forces of 0.5 N, with higher forces increasing deformations only moderately. Lateral sliding elicited strong tension and compression, often aligned with the surface orientation of the skin, which was especially strong during stick-to-slip transitions. Contact with edges and grooves yielded strong vertical shear, with ridge flanks moving relative to each other. We observed horizontal shear during static contact and sometimes when sliding the flat surface, however, it was limited in magnitude and was present only for some participants.

### Canonical ridge deformations during tactile interactions

One of the main objectives of our study was to investigate the primary deformation patterns exhibited by individual ridges during tactile interactions. Previous studies combining mechanical measurements at the ridge surface and histology (*Cauna, 1954*) or analyzing the spatial keratin expression patterns within single ridges (*Swensson et al., 1998*) have suggested the presence of alternating softer and stiffer regions within the ridges. These findings implied a mechanically constrained set of canonical deformations, yet the specific nature of these deformations has been a subject of debate, lacking direct empirical evidence.

In the present study, we have provided conclusive evidence that ridge peaks are not rigid and, instead, individual ridges undergo considerable flattening during contact. This finding challenges previous speculation that the grooves between the papillary ridges act as 'hinges' (*Swensson et al., 1998*). Instead, our results suggest that the fingerprint predominantly conforms to an external object through vertical shearing. Therefore, we propose that the primary mechanism through which a ridge conforms to an object might involve the relative movement and shearing of the ridge flanks, rather than relying on the grooves as articulated joints.

In contrast to the prominent vertical shearing observed, our study revealed minimal and inconsistent horizontal shear. We propose that the flattening of the ridges during contact may prevent further bending of the ridges, thus limiting the occurrence of horizontal shear. The absence of horizontal shear during the stick phase of lateral sliding on the flat surface was particularly unexpected. During these periods, the skin re-adhered to the plate and was subsequently dragged along, often over a distance of several millimeters. It is possible that the majority of the surface movement of the skin was absorbed by deeper tissues rather than the surface layers of the skin imaged in the present study. If that is the case, recent modeling work has suggested that tissue deformations are highly dependent on the orientation of collagen fibers in these tissues (*Duprez et al., 2024*), which might be amenable to tracking in future OCT work to test this idea directly. Additionally, previous work investigating tactile afferent responses to tangential skin movements has reported strong activation of SA-2 receptors, thought to measure skin stretch mainly in deeper tissues (*Saal et al., 2025*), providing further indirect evidence.

### Implications for neural coding of touch

Ridge deformations are ultimately transduced into neural activity by mechanoreceptors, and our findings provide new insights into this process.

First, the magnitude of sub-surface deformation measured in our study correlated with neural response strengths observed in other studies. In trials with the flat surface, we measured larger

deformations during stick-to-slip transitions than during reversals of the plate's movement direction. Analogously, a previous study recorded neural responses during sliding stimulation of the fingertip and reported that rapidly adapting type I afferents (RA-I) showed little to no response during the onset and offset of the tangential load but strongly encoded the stick-to-slip transition (*Delhaye et al., 2021*), suggesting that mechanoreceptors directly encode the local amount of deformation. Our findings also extend these prior results by demonstrating that the large surface strains measured during the transition from stick to slip are mirrored in tissues below the surface and specifically at the depth at which type-1 mechanoreceptors are located. Altogether, these results provide a physiological explanation for the widely observed human capacity of fine-tuned grasping with quick adjustments of the grip force to friction changes (*Johansson and Westling, 1984*; *Cadoret and Smith, 1996*; *Delhaye et al., 2024*).

Second, our finding that contact with small tactile features elicits shearing of both ridge flanks against each other suggests that the mechanical resolution of the fingerprint skin is on the order of half the width of an individual papillary ridge. In principle, such fine-grained and local differences in strain might be detectable by mechanoreceptors. Indeed, Meissner corpuscles innervate each flank of a fingerprint's ridge, and prior work has shown that separate fibers can innervate the two flanks of the same ridge (*Nolano et al., 2003*). Previous psychophysical work also appears to support this idea because it has estimated the spatial resolution of static touch at around 200 m, thus approximately half the width of a single ridge of the fingerprint (*Hollins and Risner, 2000*; *Bensmaia and Hollins, 2003*). However, it should be pointed out that such estimates are notoriously difficult to obtain reliably as they depend on both the precise stimulus and the mode of contact and that this limit has traditionally been associated with the responses of Merkel cells rather than Meissner corpuscles. Previous neural recordings measured the spatial acuity of the receptive sub-fields of RA1 and SA1 afferents, which were estimated to lie in the submillimeter range, roughly matching the width of a single ridge (*Jarocka et al., 2021*; *Sukumar et al., 2022*). Our results also suggest that complementary features (edges and grooves, respectively) generate symmetrical deformation. Recording responses from the same tactile afferent to such complementary features might help to disentangle what aspects of shearing the afferent is sensitive to. Therefore, further electrophysiological and psychophysical work is needed to determine whether the sub-ridge mechanical resolution of the fingerprint skin does indeed translate into corresponding neural and perceptual acuity.

## Limitations and future work

While our study was based on a large dataset, which contained a variety of tactile interactions, there are a number of limitations. First, while we could confidently track landmarks associated with the stratum corneum, we could not reliably identify intermediate ridges in the viable epidermis, though they were visible in some of the frames, limiting the depth of the fitted mesh. We hypothesize that the additional depth of these ridges, combined with their slender morphology, might have degraded the signal. 3D OCT imaging (see below) might help to resolve these features in future work and settle open questions regarding their precise morphology. Intermediate ridges are of importance because Merkel cell clusters are located at their tip, and they are, therefore, likely important for mechanotransduction. Indeed, intermediate ridges feature in the so-called 'lever hypothesis' of tactile transduction (*Cauna, 1954*), which has been called into question (*Gerling and Thomas, 2005*; *Gerling and Thomas, 2008*), though direct empirical evidence is currently lacking. Ideally, future studies would improve imaging resolution and clarity to visualize morphological structures in more detail and at greater depth. We employed a state-of-the-art, clinical OCT scanner, routinely used in dermatological clinical assessments. Data from a recently developed experimental scanner demonstrated the possibility of resolving smaller structures in the tissues, even as small as Meissner corpuscles, by applying a technique called speckle-modulating OCT (*Liba et al., 2017*). Coupling such advanced imaging techniques with our experimental setup and analysis pipeline would further advance the understanding of sub-surface skin properties and deformation mechanisms and allow more direct investigation of neural mechanisms. Such advanced imaging could also be combined with electrophysiological recordings from afferent fibers using microneurography (see *Delhaye et al., 2021*, for an example).

Second, while the edges and grooves of our feature plates were only marginally larger than a single papillary ridge and mimic those employed in other studies (*Jarocka et al., 2021*), even smaller stimuli would be capable of delivering much more localized forces (*Johansson and LaMotte, 1983*;

*Kao et al., 2022*). In the present study, we were limited by the resolution of the 3D printing technique employed to manufacture the transparent plates. Additionally, sharp corners elicit optical artifacts in the obtained images, which can occlude landmarks.

Third, we obtained two-dimensional slices oriented orthogonal to the orientation of the fingerprints, but the skin is a three-dimensional structure. Some of our tactile interactions might have caused skin deformations out of plane that were thus not measurable. For example, the slight increase in thickness of the stratum corneum under normal load might be explained as a measurement artifact due to the coarse nature of the mesh fitted, but could alternatively reflect tissue from out-of-plane regions pushing into the imaged space. Indeed, recent surface measurements of the skin's behaviour during initial object contact have reported compression of the skin in the plane parallel to its surface (*Doumont et al., 2025*), which would result in increasing thickness, assuming that the stratum corneum is incompressible. Future studies could consider creating three-dimensional reconstructions of the fingerprint structure to study such effects.

Fourth, in the present study, we focused on young, healthy participants, who were tested in a temperature and humidity-controlled environment. However, several studies have shown that morphological and mechanical skin properties can greatly vary with physiological and physical factors such as age (*Jobanputra et al., 2020*; *Püllen et al., 2021*; *Skedung et al., 2018*), sex (*Luebberding et al., 2014*; *Diridollou et al., 2000*), skin hydration (*Tomlinson et al., 2011*), moisturization (*Dione et al., 2023*), as well as ambient temperature and humidity levels (*Klaassen et al., 2016*). All these factors might, therefore, also influence how individual ridges deform and present opportunities for future studies.

Finally, while we focused on the fingertip only, many other skin regions present interesting mechanical challenges waiting to be explored. The general ridged structure observed on the fingertip is common to all glabrous skin, but the local ridge mechanics might still differ: glabrous skin on the foot sole exhibits some morphological differences in order to support large weights that might well influence its mechanical response (*Boyle et al., 2019*). For example, the morphology of transverse ridges (running orthogonal to and connecting limiting with intermediate ridges) differs across regions on the foot sole (*Nagashima and Tsuchida, 2011*) and very likely from the hand (*Yamada et al., 1996*). Our method should be directly applicable to study deformations of these ridges, though three-dimensional observations might be needed to resolve some of the open questions. Hairy skin, in contrast, differs from glabrous skin in that the stratum corneum is much thinner. It also lacks the clearly organised ridge structure but exhibits more loosely oriented skin folds instead, which very likely also serve a mechanical function (*Leyva-Mendivil et al., 2015*) and in principle are amenable to study using OCT.

## Materials and methods

### Participants

Ten healthy participants (3 males, average age 20.5 years) participated in the experiment after providing informed consent. Due to low imaging quality for data from one participant, which precluded consistent tracking, all sliding experiments included data from nine participants; the static protocol was run on a subset of five participants. The experiment was performed on the subjects' left middle finger. Before any trial, the finger was thoroughly cleaned with water and dried carefully, and a thin layer of petroleum jelly was applied to moisturize the skin; this helped prevent repeated stick-slip events while sliding the surface. Environmental conditions during the test were controlled with 20°C room temperature and 50% relative humidity. The study protocol received ethical approval from The University of Sheffield Research Ethics Committee (ethics number 039144). All experiments took place in the Skin Barrier Research Facility, operated by Sheffield Dermatology Research at the Royal Hallamshire Hospital in Sheffield, UK.

### Experimental setup

The experimental setup for the present study was adapted from the design of an earlier study (*Lee et al., 2020*). Transparent plates were secured by a support rig fixed on a linear stage that could move in both distal and proximal directions with respect to the participant's fingertip, allowing the plate to slide against the finger. The fingertip was glued onto a finger holder to maintain its position during the acquisitions. The support rig was mounted on a force plate (HE6X6–10, Advanced Mechanical

Technology, Inc) to measure the load along the axes parallel and perpendicular to the direction of the movement and along the axis normal to the plate surface. The plates were made from polymethyl methacrylate (PMMA) and were 30 × 40 × 0.5 mm in size (produced by Shape Technology S.r.l., Casale Monferrato, Italy). One plate was flat, while the two others had an embossed oriented half-circular edge (1 mm base diameter and 0.4 mm height) or an engraved oriented groove (1 mm base diameter and 0.3 mm depth) traversing the middle of the surface (see *Figure 1F*).

Prior to each trial, the plate specimens were cleaned with deionized water and dried thoroughly with paper towels. The participant's finger was secured in a finger holder, which was positioned in such a way that the flat part of the fingertip distal to the whorl made initial contact with the plate as it was lowered onto the fingertip. The scanner was positioned such that its scan path aligned with the distal-proximal axis of the plate, targeting the centre line of the fingerpad so that the fingerprint ridges were oriented orthogonally to the line scan. In the static loading experiment, the plate was initially positioned in contact with the participant's fingertip without exerting any pressure (~0 N). Afterward, the load was manually incrementally increased in 0.5 N steps, reaching a maximum of 3.5 N. At each step, 20 frames were recorded. In the sliding experiments, before commencing the imaging acquisition in each trial, the plate was manually lowered onto the participant's fingertip until a normal load of 0.2 N was reached. During each trial, the plate was moved 7.6 mm in each direction (proximal to distal and vice versa) four times at a constant speed of 0.8 mm/s. For these experiments, imaging focused on the central flat part of the contact area, such that all fingerprint ridges visible in the imaged region were in contact with the plate throughout the trial.

## Image acquisition and pre-processing

For image acquisition, we utilized the Vivosight OCT system (Michelson Diagnostics, Kent), which is clinically approved and features a Fourier domain with a 20 kHz swept-source laser at 1300 nm center wavelength. The image capture rate was 10 frames per second, and the dimension of each image was 895 × 483 pixels with 256 gray levels with a 4.5 μm lateral and 5 μm axial resolution.

Following the acquisition, we preprocessed the images using the cv2 library in Python. Firstly, we normalized the brightness and increased the contrast by applying histogram equalization. Then, to further enhance the contrast and adjust the image saturation, we performed a gamma transformation ($\gamma = 5$). Lastly, we removed noise while preserving sharp edges by applying a bilateral filter (diameter of pixel neighborhood: 5, sigmaColor: 110, sigmaSpace: 190).

## Tracking of individual ridge deformation

In each frame of the experiment, the top and bottom of each ridge within the field of view were semi-automatically tracked at three levels: the surface, the border between the stratum corneum and viable epidermis, and the junction between the dermis and epidermis. As imaging slices were 4 mm wide, often 8–9 ridges were visible concurrently. For each set of frames obtained during a single trial, a portion of the frames, ranging between 25–50% of the total, were manually annotated using the Python polygonal annotation library Labelme (*Wada, 2021*). The annotated frames were then used to train a DeepLabCut model to automatically estimate the position of each tracked point in the remaining frames (*Nath et al., 2019*). The accuracy of the models, on average, was 2.96 pixels on the training set and 14.64 pixels on the test set. We then manually reviewed the automatic annotations to correct any inaccuracies that might have arisen from automatic tracking. To measure the deformation of individual sub-surface ridges, the tracked points were used to create a triangular mesh for each ridge in the field of view. The mesh consisted of 8 facets per ridge, divided between the stratum corneum and viable epidermis and between the proximal and distal flanks of each ridge.

## Movement phase classification

To study the skin movement during the transits of the plate, we analyzed the displacement of the tracked points from frame to frame. We calculated the horizontal and vertical components of the skin velocity, $v_x$ and $v_y$, respectively, by measuring the displacement along the x-axis and y-axis. To determine the velocity components for each ridge, we averaged the velocities of the nine tracked points belonging to that ridge, resulting in $v_x^{ridge}$ and $v_y^{ridge}$.

For the flat plate trials, four phases were identified by analyzing the horizontal component of the velocity $v_x^{ridge}$:

1. Sticking proximally ($v_x^{plate} < 0$, $v_x^{ridge} < 0$): the plate is moving in distal to proximal direction, and the skin is sticking to the plate.
2. Slipping proximally ($v_x^{plate} < 0$, $v_x^{ridge} = 0$): the plate is moving in distal to proximal direction, and it is slipping over the stationary skin.
3. Sticking distally ($v_x^{plate} > 0$, $v_x^{ridge} > 0$): the plate is moving in proximal to distal direction, and the skin is sticking to the plate.
4. Slipping distally ($v_x^{plate} > 0$, $v_x^{ridge} = 0$): the plate is moving in proximal to distal direction, and it is slipping over the stationary skin.

To identify the transition between sticking and slipping phases, we took the positive and negative extrema of the first derivative of $v_x^{ridge}$ as markers. Each trial consisted of four plate transits in each direction, resulting in a total of 16 successive phases. We selected the central $n_{phase}$ frames for each phase to ensure an equal number of frames for each occurrence of a single phase during the subsequent mesh averaging. We chose $n_{phase}$ as the minimum duration of each phase among its four repetitions. The time spent in either phase varied markedly across participants (range: 11.4–82.7%, mean: 44.7%). These differences are partly attributable to differences in skin mechanics across participants (e.g. moisture level) but also depend on the precise location of the imaging site on the fingertip, as previous research has shown that stable contact is lost progressively over time across the whole fingertip during the transition from stick to slip (*Delhaye et al., 2014*).

For trials using the feature plates, all frames where $v_x^{ridge} \neq 0$ were discarded to retain only data where the skin was fully slipping under the plate. Then, by examining the vertical component of the skin speed $v_y^{ridge}$, four phases were classified for every ridge mesh separately:

1. full slipping ($v_y^{ridge} = 0$): before and after approaching the feature
2. approaching the feature ($v_y^{ridge} > 0$ for the groove plate and $v_y^{ridge} < 0$ for the edge plate)
3. moving away from the feature ($v_y^{ridge} < 0$ for the groove plate and $v_y^{ridge} > 0$ for the edge plate)
4. under the plate feature (frames in between the two previous phases, $v_y^{ridge} = 0$)

In each trial, there were a total of 40 successive phases, as each of the five phases identified occurred in each of the eight transits of the plate. Again, the central $n_{phase}$ frames were selected for each phase in order to consider the subsequent mesh averaging an equal number of frames in each occurrence of a single phase.

## Strain computation

The mesh coordinates of each ridge in each frame were centered around the origin by subtracting from the coordinates of any tracked point in a ridge, the average of the coordinates across all the points in that ridge. Then, to obtain a single stereotypical ridge shape for every phase, the mesh coordinates of each ridge were averaged across all the ridges in a frame and all the frames in the same phase. Displacement field gradients were calculated for each triangle in the mesh,

- between two consecutive phases for the flat plate trials
- between the initial slipping phase and every other phase for the feature plates.

Green-Lagrange strains were estimated from the displacement gradient by the equations:

$$\varepsilon_{xx} = \frac{\partial u}{\partial x} + 0.5 \left[ \left( \frac{\partial u}{\partial x} \right)^2 + \left( \frac{\partial v}{\partial x} \right)^2 \right] \tag{1}$$

$$\varepsilon_{yy} = \frac{\partial v}{\partial y} + 0.5 \left[ \left( \frac{\partial u}{\partial y} \right)^2 + \left( \frac{\partial v}{\partial y} \right)^2 \right] \tag{2}$$

$$\varepsilon_{xy} = 0.5 \left( \frac{\partial u}{\partial y} + \frac{\partial v}{\partial x} \right) + 0.5 \left( \frac{\partial u}{\partial x} \frac{\partial u}{\partial y} + \frac{\partial v}{\partial x} \frac{\partial v}{\partial y} \right) \tag{3}$$

where $\varepsilon_{xx}$ and $\varepsilon_{yy}$ are the horizontal and vertical strain, $\varepsilon_{xy}$ is the shear strain, and $u$ and $v$ are the displacements along x and y axes, respectively.

The principal components of the strain, $e_1$ and $e_2$, were obtained by eigenvalue decomposition of the strain matrix $\varepsilon$, as reported in *Delhaye et al., 2016*. The principal strain decomposition consists of a rotation of the reference coordinates so that the shear strain is canceled and the axial strains take

their maximal and minimal value. Thus, the principal components $e_1$ and $e_2$ represent the maximum tensile and compressive deformation along perpendicular axes.

From the principal strain components, we computed the variation in the area of every triangle composing the mesh as:

$$e_a = \frac{e_1 + e_2}{2} \tag{4}$$

As a measure of overall deformation during sliding of the flat surface, we calculate the maximum shear strain as:

$$e_s = \frac{e_1 - e_2}{2} \tag{5}$$

## Statistics

Statistical tests were run in Python using the scipy.stats package. As distributions were skewed, we used non-parametric analyses throughout the study. Bonferroni corrections were used when multiple comparisons were made.

## Acknowledgements

We would like to thank Simon Danby and Paul Andrew for making the Skin Barrier Research Facility available and Mia Rupani for assistance with data processing.

## Additional information

### Funding

| Funder | Grant reference number | Author |
| --- | --- | --- |
| Horizon 2020 Framework Programme | 10.3030/813713 | Giulia Corniani Hannes P Saal |
| Engineering and Physical Sciences Research Council | EP/R001766/1 | Zing S Lee Matt J Carré Roger Lewis |
| Fonds De La Recherche Scientifique - FNRS | | Benoit P Delhaye |

The funders had no role in study design, data collection and interpretation, or the decision to submit the work for publication.

### Author contributions

Giulia Corniani, Conceptualization, Data curation, Software, Formal analysis, Investigation, Visualization, Methodology, Writing – original draft, Writing – review and editing; Zing S Lee, Conceptualization, Resources, Software, Methodology; Matt J Carré, Roger Lewis, Conceptualization, Resources, Funding acquisition, Writing – review and editing; Benoit P Delhaye, Conceptualization, Software, Formal analysis, Supervision, Methodology, Writing – review and editing; Hannes P Saal, Conceptualization, Data curation, Supervision, Funding acquisition, Methodology, Writing – original draft, Writing – review and editing

### Author ORCIDs

Giulia Corniani ⓘ https://orcid.org/0000-0003-2800-3329
Matt J Carré ⓘ https://orcid.org/0000-0003-3622-990X
Benoit P Delhaye ⓘ https://orcid.org/0000-0003-3974-7921
Hannes P Saal ⓘ https://orcid.org/0000-0002-7544-0196

### Ethics

Ten healthy participants (3 males, average age 20.5 years) participated in the experiment after providing written informed consent, including consent to publish the recorded data. The study

protocol received ethical approval from The University of Sheffield Research Ethics Committee (ethics number 039144).

Reviewer #2 (Public review): https://doi.org/10.7554/eLife.93554.3.sa1
Reviewer #3 (Public review): https://doi.org/10.7554/eLife.93554.3.sa2
Author response https://doi.org/10.7554/eLife.93554.3.sa3

---

# Additional files

## Supplementary files
MDAR checklist

## Data availability
All OCT data and main analysis code are available at Dryad, https://doi.org/10.5061/dryad.0vt4b8hbz.

The following dataset was generated:

| Author(s) | Year | Dataset title | Dataset URL | Database and Identifier |
|---|---|---|---|---|
| Corniani G, Lee ZS, Carré MJ, Lewis R, Delhaye B, Saal HP | 2025 | Data and code from: Sub-surface deformation of individual fingerprint ridges during tactile interactions | https://doi.org/10.5061/dryad.0vt4b8hbz | Dryad Digital Repository, 10.5061/dryad.0vt4b8hbz |

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
