## [Editor Report · eLife Assessment]

By leveraging optical coherence tomography this study provides **important** insight into the deformation of human fingertip ridges when contacting raised features such as edges and contours. The study provides **compelling** evidence that such features tend to cause deformation and relative movement of what the authors term ridge flanks rather than bending of the ridges themselves.

---

## [Referee Report · Reviewer #2 (Public review)]

Summary:

The authors investigate sub-skin surface deformations to a number of different, relevant tactile stimuli, including pressure and moving stimuli. The results demonstrate and quantify the tension and compression applied from these types of touch to fingerprint ridges, where pressure flattens the ridges. Their study further revealed that on lateral movement, prominent vertical shearing occurred in ridge deformation, with somewhat inconsistent horizontal shear. This also shows how much the deeper skin layers are deformed in touch, meaning the activation of all cutaneous mechanoreceptors, as well as the possibility of other deeper non-cutaneous mechanoreceptors.

Strengths:

The paper has many strengths. As well as being impactful scientifically, the methods are sound and innovative, producing interesting and detailed results. The results reveal the intricate workings of the skin layers to pressure touch, as well as sliding touch over different conditions. This makes it applicable to many touch situations and provides insights into the differential movements of the skin, and thus the encoding of touch in regards to the function of fingerprints. The work is very clearly written and presented, including how their work relates to the literature and previous hypotheses about the function of fingerprint ridges. The figures are very well-presented and show individual and group data well. The additional supplementary information is informative and the video of the skin tracking demonstrates the experiments well.

Weaknesses:

There are very few weaknesses with the work; rather the authors detail well the limitations in the discussion. Therefore, this opens up lots of possibilities for future work.

Impact/significance:

Overall, the work will likely have a large impact on our understanding of the mechanics of the skin. The detail shown in the study goes beyond current understanding, to add profound insights into how the skin actually deforms and moves on contact and sliding over a surface, respectively. The method could be potentially applied in many other different settings (e.g. to investigate more complex textures, how skin deformation changes with factors like dryness and aging). This fundamental piece of work could therefore be applied to understand skin changes and how these impact on touch perception. It can further be applied to understand skin mechanoreceptor function better and model these. Finally, the importance of fingertip ridges is well-detailed, demonstrating how these play a role in directly shaping our touch perception and how they can shape the interactions we have with surfaces.

---

## [Referee Report · Reviewer #3 (Public review)]

Summary:

The publication presents unique in-vivo images of the upper layer of the epidermis of glabrous skin when a flat object compresses or slides on the fingertip. The images are captured using OCT and show the strain that fingerprints experience during mechanical stimulation.

The most important finding is, in my opinion, that fingerprints undergo pure compression/tension without horizontal shear, suggesting that the shear stress caused by tangential load is transferred to the deeper tissues and ultimately to the mechanoreceptors (SA-I / RA-I).

Strengths:

Fascinating new insights into the mechanics of glabrous skin. To the best of my knowledge, this is the first experimental evidence of the mechanical deformation of fingerprints when subjected to dynamic mechanical stimulation. The OCT measurement allows unprecedented measurement of skin depth, whereas previous works were limited to tracking surface deformation.

The robust data analysis reveals the continuum mechanics underlying the deformation of the fingerprint ridges.

Weaknesses:

I do not see any major weaknesses. The work is mainly experimental and is rigorously executed.

---

## [Author Response]

The following is the authors’ response to the original reviews.

**Reviewer #1 (Public Review):**
Summary:This manuscript uses optical coherence tomography (OCT) to visualize tissue microstructures about 1-2 mm under the finger pad skin surface. Their geometric features are tracked and used to generate tissue strains upon skin surface indentation by a series of transparent stimuli both normal and tangential to the surface. Then movements of the stratum corneum and the upper portion of the viable epidermis are evaluated. Based upon this data, across a number of participants and ridges, around 300 in total, the findings report upon particular movements of these tissue microstructures in various loading states. A better understanding of the mechanics of the skin microstructures is important to understand how surface forces propagate toward the locations of mechanoreceptive end organs, which lie near the edge of the epidermis and dermis, from which tactile responses of at least two peripheral afferents originate. Indeed, the microstructures of the skin are likely to be important in shaping how neural afferents respond and enhance their sensitivity, receptive field characteristics, etc.Strengths:The use of OCT in the context of analyzing the movements of skin microstructures is novel. Also novel and powerful is the use of distinct loading cases, e.g., normal, tangential, and stimulus features, e.g., edges, and curves. I am unaware of other empirical visualization studies of this sort. They are state-of-the-art in this field.Moreover, in addition to the empirical imaging observations, strain vectors in the tissues are calculated over time.Weaknesses:The interpretation of the results and their framing relative to the overall hypotheses/questions and prior works could be articulated more clearly. In particular, the major findings of the manuscript are in newly describing a central concept regarding "ridge flanks," but such structures are neither anatomically nor mechanistically defined in a clear fashion. For example, "... it appears that the primary components of ridge deformation and, potentially, neural responses are deformations of the ridge flanks and their relative movement, rather than overall bending of the ridges themselves." From an anatomical perspective, I think what the authors mean by "ridge flanks" is a differential in strain from one lateral side of a papillary ridge to the other. But is it unclear what about the continuous layers of tissue would cause such behaviors. Perhaps a sweat duct or some other structure (not visible to OCT) would subdivide the "flanks" of a papillary ridge somehow? If not due to particular anatomy, then is the importance of the "ridge flank" due to a mechanistic phenomenon of some sort? Given that the findings of the manuscript center upon the introduction of this new concept, I think a greater effort should be made to define what exactly are the "ridge flanks." It is clear from the results, especially the sliding case, that there is something important that the manuscript is getting at with this concept.

We apologize for the confusion around our use of ‘ridge flanks’. To recap the overall goal briefly, we wanted to measure the deformation of papillary ridges and their associated sub-surface structures to different tactile stimuli. Capturing these deformations and comparing them against different proposed ideas, for example bending (horizontal shear) of the entire ridge versus differential deformations of different sub-parts, constrains neural activation mechanisms, has implications for how well tactile stimuli can be spatially resolved on the skin, and for whether sub-surface deformations can be easily predicted from surface movements alone. Our mesh was dense enough to compare the stratum corneum and the viable epidermis directly, where we expected some differences due to their previously documented mechanical differences, as well as the ridge flanks, which refers to the two (proximal and distal) sides of a single papillary ridge and their associated structure in the SC and VE (as correctly surmised by the reviewer). Differential behaviour across ridge flanks might be seen, because various observations of the surface of the stratum corneum had suggested mechanical differences between the papillary ridges and the grooves dividing them, potentially leading to differential deformations of these two halves depending on which direction they were facing tissue with different mechanical properties.

We now provide a clearer definition of ridge flanks in Figure 1 and in the main text. Importantly, existing prior research is better connected to our own investigation in the Introduction and we now specifically explain why we investigate ridge flanks.

The OCT used herein cannot visualize deep and fully into what the manuscript refers to as a "ridge"(note others have previously broken apart this concept apart into "papillary", "intermediate" and "limiting" ridges) near locations of the mechanoreceptive end organs lie at the epidermal-dermal border. Therefore, the OCT must make inferences about the movements of these deeper tissues, but cannot see them directly, and it is the movements of these deeper tissues that are likely driving the intricacies of neural firing. Note the word "ridge" is used often in the manuscript's abstract, introduction, and discussion but the definition in Fig. 1 and elsewhere differs in important ways from prior works of Cauna (expert in anatomy). Therefore, the manuscript should clarify if "ridge" refers to the papillary ridge (visible at the exterior of the skin), intermediate ridge (defined by Cauna as what the authors refer to as the primary ridge), and limiting ridge (defined by Cauna as what the authors refer to as the secondary ridge). What the authors really mean (I think) is some combination of the papillary and intermediate ridge structures, but not the full intermediate ridge. The manuscript acknowledges this in the "Limitations and future work" section, stating that these ridges cannot be resolved. This is important because the manuscript is oriented toward tracking this structure. It sets up the narrative and hypotheses to evaluate the prior works of Cauna, Gerling, Swensson, and others who all directly addressed the movement of this anatomical feature which is key to understanding ultimately how stresses at these locations might move the peripheral end organs (i.e., Merkel cells, Meissner corpuscles).

Thank you for these observations. Indeed, our terminology was not consistent. We have now switched to Cauna’s terminology and added additional labels in Figure 1, explaining all mentioned structures in the main text. We have also changed the language in many instances in the main text to make it clearer whether we are referring to individual anatomical ridges (papillary, limiting, etc.) or the whole structure. Additionally, it is now clearer from the start which features are tracked, and we specifically state that intermediate ridges are excluded from our tracking.

Regarding the intermediate ridge, it indeed plays a big role in Cauna’s lever hypothesis. Given the intermediate ridge is excluded from our analysis, we can neither prove nor disprove this hypothesis in our current work. However, there are many mechanical mysteries to solve regarding the structures directly above, which are the main focus of this paper. We have rewritten the introduction to make these questions clearer. For example, Cauna observed pliability of the papillary ridges in surface experiments. Swensson found differential expression patterns of keratin in epidermis tissue in and above the intermediate ridges, but the direct mechanical consequences that are proposed in their paper concern the behaviour of papillary ridges, rather than relying on a mechanical role of intermediate ridges. Even Cauna’s lever idea implies specific deformation of the stratum corneum, which would be measurable in our study, as the upper handle of the ‘lever’ needs turning. We observed little movement in accordance with this idea, putting the lever mechanism into question. While this does not rule out a mechanical role of the intermediate ridge, these findings constrain its potential mechanisms.

**Reviewer #2 (Public Review):**
Summary:The authors investigate sub-skin surface deformations to a number of different, relevant tactile stimuli, including pressure and moving stimuli. The results demonstrate and quantify the tension and compression applied from these types of touch to fingerprint ridges, where pressure flattens the ridges. Their study further revealed that on lateral movement, prominent vertical shearing occurred in ridge deformation, with somewhat inconsistent horizontal shear. This also shows how much the deeper skin layers are deformed in touch, meaning the activation of all cutaneous mechanoreceptors, as well as the possibility of other deeper non-cutaneous mechanoreceptors.Strengths:The paper has many strengths. As well as being impactful scientifically, the methods are sound and innovative, producing interesting and detailed results. The results reveal the intricate workings of the skin layers to pressure touch, as well as sliding touch over different conditions. This makes it applicable to many touch situations and provides insights into the differential movements of the skin, and thus the encoding of touch in regards to the function of fingerprints. The work is very clearly written and presented, including how their work relates to the literature and previous hypotheses about the function of fingerprint ridges. The figures are very well-presented and show individual and group data well. The additional supplementary information is informative and the video of the skin tracking demonstrates the experiments well.Weaknesses:There are very few weaknesses in the work, rather the authors detail well the limitations in the discussion. Therefore, this opens up lots of possibilities for future work.

We thank the reviewer for these encouraging comments.

Impact/significance:Overall, the work will likely have a large impact on our understanding of the mechanics of the skin. The detail shown in the study goes beyond current understanding, to add profound insights into how the skin actually deforms and moves on contact and sliding over a surface, respectively. The method could be potentially applied in many other different settings (e.g. to investigate more complex textures, and how skin deformation changes with factors like dryness and aging). This fundamental piece of work could therefore be applied to understand skin changes and how these impact touch perception. It can further be applied to understand skin mechanoreceptor function better and model these. Finally, the importance of fingertip ridges is well-detailed, demonstrating how these play a role in directly shaping our touch perception and how they can shape the interactions we have with surfaces.
**Reviewer #3 (Public Review):**
Summary:The publication presents unique in-vivo images of the upper layer of the epidermis of the glabrous skin when a flat object compresses or slides on the fingertip. The images are captured using OCT, and are the process of recovering the strain that fingerprints experience during the mechanical stimulation.The most important finding is, in my opinion, that fingerprints undergo pure compression/tension without horizontal shear, hinting at the fact that the shear stress caused by the tangential load is transferred to the deeper tissues and ultimately to the mechanoreceptors (SA-I / RA-I).Strengths:Fascinating new insights into the mechanics of glabrous skin. To the best of my knowledge, this is the first experimental evidence of the mechanical deformation of fingerprints when subjected to dynamic mechanical stimulation. The OCT measurement allows an unprecedented measurement of the depth of the skin whereas previous works were limited to tracking the surface deformation. - The robust data analysis reveals the continuum mechanics underlying the deformation of the fingerprint ridges.Weaknesses:I do not see any major weaknesses. The work is mainly experimental and is rigorously executed. Two points pique my curiosity, however:(1) How do the results presented in this study compare with previous finite element analysis? I am curious to know if the claim that the horizontal shear strain is transferred to the previous layer is also captured by these models. The reason is that the FEA models typically use homogeneous materials and whether or not the behavior in-silico and in-vivo matches would offer an idea of the nature of the stratum corneum.

Very few modeling studies have examined combined normal and tangential loading of the fingertip. Additionally, results are often expressed in terms of Von Mises stresses, and not deformation [1,2], making direct comparison challenging. Nevertheless, one multilayered study [3] supports our finding that the largest deformations are found in deeper tissues.

(1) Shao, F., Childs, T. H. C., Barnes, C. J. & Henson, B. Finite element simulations of static and sliding contact between a human fingertip and textured surfaces. Tribology International 43, 2308–2316 (2010).

(2) Tang, W. et al. Investigation of mechanical responses to the tactile perception of surfaces with different textures using the finite element method. Advances in Mechanical Engineering 8, (2016).

(3) Amaied, E., Vargiolu, R., Bergheau, J. M. & Zahouani, H. Aging effect on tactile perception: Experimental and modelling studies. Wear 332–333, 715–724 (2015).

(2) Was there a specific reason why the authors chose to track only one fingerprint? From the method section, it seems that nothing would have prevented tracking a denser point cloud and reconstructing the stain on a section of the skin rather than just one ridge. With such data, the author could extend their analysis to multiple ridges interaction and get a better sense of the behavior of the entire strip of skin.

We apologise for the confusion regarding this point. While in our illustration and the accompanying videos, we only show a single tracked ridge for clarity, we do indeed track all visible ridges in every frame. As imaging slices were 4 mm wide, often 8-9 ridges were visible concurrently. However, during the sliding experiments the skin was sometimes dragged along with the stimulus, causing some ridges to disappear from view for certain periods and then re-enter the frame. This would make it difficult to expand the analysis to multiple ridges, but in any case, we found neighbouring ridges to behave very consistently within a given trial, so that their mechanical behaviour (relative to the tactile feature, if any) could be averaged in the analysis.

**Reviewer #1 (Recommendations For The Authors):**
Discussion, line 213, "Thus, the primary mechanism through which the ridge conforms to the object involves the relative movement and shearing of the ridge flanks, rather than relying on the groves as articulated joints." I don't see this as definitely proven in the imaging and analysis. This could be a hypothesis to come from this work for further evaluation but is a quite strong statement not obviously supported by the evidence.

We have rephrased this statement as a proposal for further testing:

“Therefore, we propose that the primary mechanism through which a ridge conforms to an object might involve the relative movement and shearing of the ridge flanks, rather than relying on the grooves as articulated joints.”

Discussion, line 220, "Our findings strongly indicate that the majority of the surface movement of the skin was observed by deeper tissue rather than surface layers of the skin." But since there are no measurements of such tissues, or of collagen bundle tightening, etc. it is not obvious to me how this can be proven as it is not directly observable and was not modeled.

We have reworded this paragraph to be more cautious and have included potential avenues for future testing of this idea:

“It is possible that the majority of the surface movement of the skin was absorbed by deeper tissues rather than the surface layers of the skin imaged in the present study. If that is the case, recent modeling work has suggested that tissue deformations are highly dependent on the orientation of collagen fibers in these tissues (Duprez et al., 2024), which might be amenable to tracking in future OCT work to test this idea directly. Additionally, previous work investigating tactile afferent responses to tangential skin movements has reported strong activation of SA-2 receptors, thought to measure skin stretch mainly in deeper tissues (Saal et al., 2025), providing further indirect evidence.”

Figure 1, A. As noted elsewhere, there are issues with the naming of the anatomy, and there is no definition of the concept of "ridge flanks." Also, it does not indicate the depth point to which OCT can resolve.

We have updated and expanded the labels in Figure 1A to clarify the anatomy (along with changes in the text described above). Figure 1C now includes a sentence about the resolvability of features below the mesh:

“Detail view of a single OCT frame showing ridged skin structure and clear boundary between the stratum corneum and viable epidermis. A mesh covering the stratum corneum and the upper part of the viable epidermis (without the intermediate ridge) is overlaid spanning a single papillary ridge. The border between the viable epidermis and dermis is less clearly delineated, but some deeper features are resolved less well.”

The concept of a ridge flank is now illustrated in Figure 1B(i) and Figure 1B(iv), and referred to in both the caption and main text. Updated figure caption text:

“These deformations need not apply to the whole ridge structure but might affect different parts separately, e.g. via shearing in different directions across both ridge flanks as shown on the far right

(see darker shading to highlight a single ridge flank).”

Updated text in the main manuscript:

“Additionally, if there are indeed mechanical differences between papillary ridges and their neighbouring grooves at the level of the stratum corneum, this might result in differential movements of the two sides of each papillary ridge, here referred to as ridge flanks (see Figure 1B-iv, right, for a potential example).”

Note that Figure 4B also includes an illustration of this concept.

Figure 1, B. This mechanical representation does not capture the entirety of the papillary-intermediate ridge unit in question, as set up by the authors in the introduction. Also, in the caption it is not ridge deformation, but upper SC and VE deformation. And the OCT cannot resolve the whole ridge.

We have reworded the figure caption”

“Potential deformations of the tracked ridge structure, including the stratum corneum and the bulk of the viable epidermis, during tactile interactions, with arrows indicating the directions of relative deformation. [...]”

Importantly, the main manuscript text has been rewritten in the introduction section to clarify our research question and how much of the sub-surface ridge structure is tracked:

“From a mechanical standpoint, these conflicting interpretations raise the question of how the outermost two skin layers typically deform at the resolution of single papillary ridges, whether by tension, compression, or shear (see examples in Figure 1B). Additionally, such deformations might apply to individual papillary ridges and all their sub-surface structures equally, for example horizontal shearing that bends the papillary ridge in a certain direction, while levering its sub-surface aspects in the opposite direction. Conversely, individual parts of the ridge structure might deform differently. For example, the viable epidermis might deform to a different extent or in different directions due to its lower stiffness and different morphology. Additionally, if there are indeed mechanical differences between papillary ridges and their neighbouring grooves at the level of the stratum corneum, this might result in differential movements of the two sides of each papillary ridge, here referred to as ridge flanks (see Figure 1B-iv, right, for a potential example). To empirically address these questions, we employed Optical Coherence Tomography (OCT) to precisely measure the sub-surface deformation of individual fingerprint ridges in response to a variety of mechanical events. Specifically, we focused on the stratum corneum and the bulk of the viable epidermis (excluding intermediate ridges), which could be robustly resolved and tracked by our setup.”

Figure 1, C: While it is noted in the caption that the locations of the intermediate and limiting ridges, as well as the collagen bundles, are clearly visible, it is not clear to me, although the caption uses these words. This is especially the case below the orange mesh. From the picture, and because this is not labeled, it leaves it up to my interpretation, it seems like the secondary ridge (limiting) is larger than the primary (intermediate).

We have reworded the caption as follows:

“Detail view of a single OCT frame showing ridged skin structure and clear boundary between the stratum corneum and viable epidermis. A mesh covering the stratum corneum and the upper part of the viable epidermis (without the intermediate ridge) is overlaid spanning a single papillary ridge. The border between the viable epidermis and dermis is less clearly delineated.”

Indeed, while the intermediate ridge was often visible in the OCT images, its size was rather inconsistent and it could appear as larger or smaller than the limiting ridge, while in histological images it is generally shown as larger (however note that there is somewhat limited data). This difference might be due to imaging artifacts, e.g. limited visibility into the deeper tissues, might reflect individual differences between participants, or could indicate that intermediate ridges are not of a consistent height in the (out-of-plane) direction along a given ridge. We have clarified this in the Limitations section of the Discussion:

“[...] while we could confidently track landmarks associated with the stratum corneum, we could not reliably identify intermediate ridges in the viable epidermis, though they were visible in some of the frames, limiting the depth of the fitted mesh. We hypothesize that the additional depth of these ridges combined with their slender morphology might have degraded the signal. 3D OCT imaging (see below) might help to resolve these features in future work and settle open questions regarding their precise morphology.”

Figure 1, D, and E: How do these measurements compare with the literature? They seem reasonable to me based on a cursory review, but there is a need to directly compare, especially since measurements in this context with the OCT are novel and could be valuable.

We have clarified this in the main text and added more references to the existing literature:

“We measured an average ridge width of 0.47 mm across participants (Figure 1D), consistent with previous studies (Moore, 1989; Ohler and Cummins, 1942). Average skin layer thickness was 0.38 mm for the stratum corneum and 0.12 mm for the viable epidermis across our dataset (Figure 1E), again in agreement with previous studies using both in vivo imaging and ex vivo histology (Fruhstorfer et al., 2000; Lintzeri et al., 2022; Maiti et al., 2020).”

Abstract 4th sentence's structure makes me think that hundreds of individual fingerprint ridges can be tracked at the same time. Perhaps it could be tweaked to clearly indicate that hundreds were tracked between trials between participants.

We have changed the sentence to now read:

“Here, we used optical coherence tomography to image and track sub-surface deformations of hundreds of individual fingerprint ridges across ten participants and four individual contact events at high spatial resolution in vivo.”

Introduction, 1st sentence, the fingertip per se is not an organ, though the skin is an organ.

Changed the wording from “organ” to “structure”.

Introduction, 1st sentence, "... that convert skin deformations ..." Need to add word skin to be clear.

Done.

Introduction, 3rd paragraph, "Alternately, the grooves may be stiffer or less ...". In this paragraph, and this sentence in particular, Cauna is cited and the words groves and ridges are used. But this is not adequately explained. Cauna had distinct terminology, where he referred to papillary, intermediate, and limiting ridges, that exist in addition to ready ridges. It is important because the manuscript uses the word "ridges" in a non-specific way. This is done not just here but throughout the manuscript, and is central to the questions which can be addressed with OCT.

Anatomy has been better defined and more extensively labelled in Figure 1A, including labels for ‘papillary ridges’ and ‘grooves’. We have reworded this paragraph to better explain the concepts and how they relate to the subsequent analyses in the paper

“Consequently, the mechanical response of the skin below its immediate surface remains largely unknown, leading to conflicting interpretations in the literature. For instance, it has been proposed that the papillary ridges are stiffer than the neighbouring grooves (Swensson et al., 1998), which might imply that normal loading of the skin might not affect the ridges’ profile appreciably. Conversely, other observations have suggested that the grooves are relatively stiff, allowing the papillary ridges to deform considerably (Cauna, 1954; Johansson and LaMotte, 1983). However, the sub-surface consequences of this putative pliability during object contact or stick-to-slip transitions (see e.g. Delhaye et al., 2016) are unclear: the whole ridge structure might bend as proposed in Cauna’s lever mechanism (Cauna, 1954), but this view has proved controversial (see e.g. Gerling and Thomas, 2008), with direct empirical evidence lacking.”

Figure 1. Avoid red-green dots for colorblind accessibility. PMMA is not in the caption.

We have switched the colors of the mechanoreceptors in panel A to a colorblind-friendly scheme. We now also specify the material of the plates in the figure 1 caption.

Results, line 102. "... papillary ridge structure...." Is this the ridge to which is being referred?

In conjunction with the updated labeling in Figure 1A, we have updated the terminology throughout the paper to be more consistent.

Results, line 99. "We noted a small increase in the area of the strateum corneum, which was likely an artifact due to the fit of the mesh to the ridge's curvature ..." There is very little discussion of Fig. F's finding related to an increase in area in the SC and decrease in the VE. It makes me question if this finding in this panel is an artifact. With stiff tissue like stratum corneum, how would the area increase?

This finding could be a measurement artifact or it could be the result of skin from neighbouring regions pushing into the imaged space. We have reworded the brief description in the Results:

“We noted a small increase in the area of the stratum corneum, which was possibly an artifact due to the imperfect fit of the mesh to the ridge's curvature (but see Discussion for an alternative explanation).”

Additionally, we have added a short section in the Discussion in the Limitations section:

“Some of our tactile interactions might have caused skin deformations out-of-plane that were thus not measurable. For example, the slight increase in thickness of the stratum corneum under normal load might be explained as a measurement artifact due to the coarse nature of the mesh fitted, but could alternatively reflect tissue from out-of-plane regions pushing into the imaged space. Indeed, recent surface measurements of the skin's behaviour during initial object contact have reported compression of the skin in the plane parallel to its surface (Doumont et al., 2025), which would result in increasing thickness, assuming that the stratum corneum is incompressible. Future studies could consider creating three-dimensional reconstructions of the fingerprint structure to study such effects.”

Figure 3. The colors used in slip and stick are not colorblind accessible.

We have changed the background colors in Figure 3A,B,C to a colorblind accessible version.

Results, line 151, "Thus, most of this shearing must be sustained by deeper tissues." But there are no direct observations as such. Also, in the next sentence, "collagen fiber bundles" are referred to in a non-specific way. This section is highly speculative with no systematic visualization of these structures, and should probably be moved to the discussion.

We have reworded this sentence to be more cautious. We have now also highlighted collagen fiber bundles visible in the figure. Systematic analysis of these is beyond the scope of the present study, as these were not tracked, but might be possible in future studies. The reworded sentence reads as follows:

“Thus, it is possible that shearing is sustained by deeper tissues, an effect that could be tested in future studies by directly tracking the angle and orientation of collagen fiber bundles anchoring the epidermis to deeper tissues (see highlighted examples in Figure 3B).”

Results, line 161, " Horizontal shear ..." do you mean surface shear, per the Fig. 1 definition?

For consistency, we have changed the labels to ‘Horizontal shear’ and ‘Vertical shear’ in Figure 1A(iii) and Figure 1A(iv) as these are the terms used throughout the paper.

Discussion, line 198, "... flatten even at relatively low forces." This is an interesting point and it would be useful to note how low exactly.

We have reworded this sentence to better reflect the findings described earlier:

“We found that individual ridges tended to flatten considerably at relatively low forces of 0.5 N, with higher forces increasing deformations only moderately.”

**Reviewer #2 (Recommendations For The Authors):**
Minor comments that could improve the paper even furtherIn the abstract, it may be good to specify that the stimuli were all applied to the finger, this was not an active, self-generated tactile interaction, e.g. change 'in response to a variety of tactile stimuli' to 'in response to a variety of passively-applied tactile stimuli'.

Done.

Comment on the grey/blue colours in the figures. I like the combination of blue/orange for different conditions, but sometimes the blue is very difficult to see against the grey background. Is there any way of making the grey background shading lighter and/or the blue darker/more vivid?

We have changed the color of the SC mesh to a darker shade of blue, which is more easily distinguished from the grey background. This applies to figures 2B/C, 3D, 4A/B/D/E, and all supplementary figures.

Methods. Could you please add a little more detail about exactly where the images were taken, e.g. in the exact middle of the fingerpad, at the fingertip? Did you line up the skin fingerprint ridges to be in a plane? It is just to better understand how the stimulus moved against the skin, which itself is rounded, and whether it was at a point where the ridges were relatively linear or curved.

We have added the following text in the “Experimental set-up” section of the Methods:

“The participant's finger was secured in a finger holder, which was positioned in such a way that the flat part of the fingertip distal to the whorl made initial contact with the plate as it was lowered onto the fingertip. The scanner was positioned such that its scan path aligned with the distal-proximal axis of the plate, targeting the centre line of the fingerpad so that the fingerprint ridges were oriented orthogonally to the line scan.”

and

“For these experiments, imaging focused on the central flat part of the contact area, such that all fingerprint ridges visible in the imaged region were in contact with the plate throughout the trial.”

Methods. There is no section about statistics, yet you do use them in the paper. It may be good to add a few details in the methods to outline the package you used to do the statistics, as well as why you chose the tests you carried out.

We have added a new Statistics section at the end of the Methods:

“Statistical tests were run in Python using the scipy.stats package. As distributions were skewed, we used non-parametric analyses throughout the study. Bonferroni corrections were used when multiple comparisons were made.”

A very minor point. Discussion, line 210: 'In this study...' is vague, which study exactly? It is preferable to be more precise, e.g. 'In the present/current study...'.

Fixed.

Discussion. One point you may want to add is the possibility of looking at other skin regions. For example, would this approach work on the palm, on border glabrous/hairy skin, on various hairy skin sites, and on the foot? The possibilities could be endless if it could be applied anywhere, but it may depend on the technical positioning and skin itself. However, it would be interesting to know.

We have added the following text at the end of the Discussion section:

“Finally, while we focused on the fingertip only, many other skin regions present interesting mechanical challenges waiting to be explored. The general ridged structure observed on the fingertip is common to all glabrous skin, but the local ridge mechanics might still differ: glabrous skin on the foot sole exhibits some morphological differences in order to support large weights that might well influence its mechanical response (Boyle et al., 2019). For example, the morphology of transverse ridges (running orthogonal to and connecting limiting with intermediate ridges) differs across regions on the foot sole (Nagashima and Tsuchida, 2011) and very likely from the hand (Yamada et al., 1996). Our method should be directly applicable to study deformations of these ridges, though three-dimensional observations might be needed to resolve some of the open questions. Hairy skin in contrast differs from glabrous skin in that the stratum corneum is much thinner. It also lacks the clearly organised ridge structure, but exhibits more loosely oriented skin folds instead, which very likely also serve a mechanical function (Leyva-Mendivil et al., 2015) and in principle are amenable to study using OCT.”

In the last lines of the discussion, you mention the possible effects of skin moisturization. The Tomlinson et al. paper refers to the hydration of the skin with regard to water, which I would say is a slightly different factor. I think you can mention this paper and talk about the water level of the skin/hydration, but also add specifically that moisturization (i.e. by an emollient, humectant, or occlusive substance) is another factor to consider (e.g. effects found by Dione et al, 2023 Sci Rep). Overall, these two points relate to the dryness of the skin and the humidity of surfaces being contacted, therefore you could expand on both.

Thank you for the correction! We now mention both skin hydration and moisturization separately in this section.